# Functional representation of the network organisation of dialysis activities in France: A novel level for assessing quality of care

**Cécile Couchoud**[1]*, **René Ecochard**[2,3], **Mathilde Prezelin-Reydit**[4], **Thierry Lobbedez**[5], **Florian Bayer**[1], in the name of the REIN registry[¶]

**1** REIN Registry, Agence de la Biomédecine, Saint-Denis La Plaine, France, **2** Université Lyon I, CNRS, UMR 5558, Laboratoire de Biométrie et Biologie Evolutive, Equipe Biostatistique Santé, Villeurbanne, France, **3** Hospices Civils de Lyon, Service de Biostatistique, Lyon, France, **4** Maison du REIN—AURAD Aquitaine, Gradignan, France, **5** Service de Néphrologie, Centre Hospitalo-Universitaire, Caen, France

¶ Dialysis and transplant centres participating in the registry are listed in the REIN annual report (https://www.agence-biomedecine.fr/Les-chiffres-du-R-E-I-N).
* cecile.couchoud@biomedecine.fr

**Data Availability Statement:** Data are available from the REIN scientific registry Committee (contact via Dr Christian Jacquelinet christian.

## Abstract

To assess quality of care, groups of care units that cared for the same patients at various stages of end-stage renal disease, might be more appropriate than the centre level. These groups constitute "communities" that need to be delineated to evaluate their practices and outcomes. In this article, we describe the use of an agglomerative (Fast Greedy) and a divisive (Edge Betweenness) method to describe dialysis activities in France. The validation was based on the opinion of the field actors at the regional level of the REIN registry. At the end of 2018, ESRD care in France took place in 1,166 dialysis units. During 2016–2018, 32 965 transfers occurred between dialysis units. With the Edge Betweenness method, the 1,114 French dialysis units in metropolitan France were classified into 156 networks and with the Fast Greedy algorithm, 167 networks. Among the 32 965 transfers, 23 168 (70%) were defined in the same cluster by the Edge Betweenness algorithm and 26 016 (79%) in the same cluster by the Fast Greedy method. According to the Fast Greedy method, during the study period, 95% of patients received treatment in only one network. According to the opinion of the actors in the field, the Fast Greedy algorithm seemed to be the best method in the context of dialysis activity modelling. The Edge Betweenness classification was not retained because it seemed too sensitive to the volume of links between dialysis units.

## Introduction

Many studies have explored the center effect on various outcomes, thus suggesting that actions at the center level can improve the quality of care. However, in the French context, the best level for assessing quality of care is still debated, particularly in the field of end-stage renal disease. Dialysis activity is organized and regulated regionally, with transplantation activities organized at a supra-regional level. Regional health agencies (*Agence Régionale de Santé*) are in

jacquelinet@biomedecine.fr, coordinator of the registry at the Agence de la biomédecine) for researchers who meet the criteria for access to confidential data. The REIN registry has its own rule for access to individual data in accordance with the French National Commission for Data Protection and Liberties (CNIL-France).

**Funding:** The author(s) received no specific funding for this work

**Competing interests:** The authors have declared that no competing interests exist

charge of planning ambulatory and hospital care by building a regional health project based on population needs. Dialysis units are widespread throughout the country. In France, the level of the dialysis unit may be inappropriate for quality of care assessment. Indeed, different modalities of dialysis are offered, each characterized by the technique of dialysis (haemodialysis or peritoneal dialysis), the extent of professional assistance, and the treatment site (hospital-based, satellite or self-care facilities, home dialysis), regardless of the provider (public, private for-profit, private not-for-profit, university hospital) [1, 2]. Consequently, patients are transferred between the different dialysis units according to their needs and choice, although the choice is mainly according to proximity to the home.

To assess quality of care, an intermediate level, that is, groups of care units that cared for the same patients at various stages of end-stage renal disease, might be more appropriate. These groups constitute "communities" that need to be delineated to evaluate their practices and outcomes. This other level, representing informal collaborations between units, is not a priori defined and is not based on contentless declared collaboration or artificial grouping. The choice of methods for grouping dialysis units into suitable clusters for quality assessment is a current topic of study.

Computerized data allow for linking the stays of the same patient in different institutions and thus identifying the facilities taking care of the same groups of patients, the communities [3, 4]. Methods for identifying communities within networks are mainly of two types: 1) "agglomerative" [5], starting with small clusters of facilities transferring many patients and gradually increasing the size of the clusters, adding facilities with this criterion, and 2) "divisive" [6], starting with all facilities in one cluster and gradually dividing the cluster into sub-groups of smaller size but ones that transfer more patients. In both cases, the rule is based on the number of patients exchanged between 2 facilities. Roughly speaking, in the agglomerative methods, the question asked at each step is which other facility the members of the agglomerate can exchange more patients with. In the divisive algorithm, the question asked at each step is how to cut the cluster in 2 by cutting where the number of patients exchanged is higher.

In this article, we describe the use of an agglomerative and a divisive method to describe dialysis activities in France and compare the results. We discuss the 2 clustering methods from a theoretical point of view.

## Methods

### Source of information

The Renal Epidemiology and Information Network (REIN) is the French national registry of all patients receiving renal replacement therapy (RRT) [7]. Clinical, demographic, and laboratory data are collected at RRT initiation, as are dialysis modalities, and are updated annually. Events such as death, transfer, dialysis withdrawal, placement on a transplant wait list, and kidney transplantation (from living or deceased donors) are systematically reported in real time. Short-term transfers, shorter than 2 months (i.e., hospital stay, vacation), are not recorded. At the regional level, a coordinating nephrologist and clinical research assistants working with a public health department for methodological support maintain the dynamics of data collection and ensure data quality control.

The French adult dialysis units are widely dispread around the country (more than 1 150 dialysis units as of December 2018), with various numbers of patients (median 25, interquartile range [IQR] 11–59, from 2 up to 320). In metropolitan France, the median time to a dialysis unit is 17 min.

Approvals from the National Commission on Informatics and Liberty and from the Advisory Committee on Information Processing in Material Research in the Field of Health were

obtained through the national REIN registry. The patients have an opt out option if they don't want to be included into the registry and the patients associations are participating to the monitoring of the registry.

## Vocabulary

Describing the methods used to group dialysis units into communities requires applying the language of social networks to that of care networks.

A network of healthcare facilities consists of vertices or nodes (the facilities) and edges (patient transfers) between them. The shortest path between 2 institutions is 1 if the institutions exchange patients with each other, with one edge between them. Otherwise, it is 2 if a third institution is between them (2 edges, 3 institutions), it is 3 for 4 institutions and 3 for shortest-path edges etc. This distance should not be considered a geographical distance (i.e., in kilometres), but as a connectivity distance.

A healthcare facility may be described by its degree (i.e., the number of facilities with which it shares patient care [from 0 to n-1]). The index of a facility's centrality is based on the average degree of proximity to all other facilities in the community [8]. A healthcare facility of high degree can be of high closeness centrality. The closeness centrality of a facility measures its average "farness" (inverse distance) to all other facilities. Typically, a referral hospital that receives patients from a large number of other institutions has a high closeness centrality. Even if 2 facilities are not directly linked to each other and do not transfer patients between them, they are indirectly linked if both exchange patients with the same facility, such as the referring facility. A healthcare centre with high closeness centrality may have more influence on the medical practices of the community than one with low closeness centrality.

The number of indirect links that pass through this facility are called the healthcare facility "betweenness". A facility with high closeness centrality is at the origin of a high healthcare facility betweenness.

A healthcare facility that exchanges patients with a large number of other facilities has a "high degree of proximity". This healthcare facility has a central position in the community. This results in a large number of indirect links between other institutions: other institutions exchange patients with this same institution having a central position in the community. This is called "high betweenness". The "distance" between this central institution and other institutions is said to be small. Therefore, high degree and high betweenness are two factors of centrality of an institution in a community.

At the opposite extreme is a facility that exchanges patients with few other facilities. Few paths (low betweenness) connecting the facilities will pass through these outlying facilities. This care facility is likely considered peripheral in the community of care. The "distance" between it and other institutions is said to be high [8].

Facility clusters are groups of facilities (modules or communities) that share more patients with facilities in the same cluster than with facilities in other clusters. *Modularity* measures when the division is a good one, in the sense that there are many transfers within communities and only a few between them. The modularity index is illustrated in Fig 1.

Skewness is a measure of asymmetry observed in a probability distribution that deviates from the symmetrical normal distribution in a given set of data. The distribution may be positively (the mean of the data is greater than the median) or negatively (the mean of the data is less than the median) skewed.

Kurtosis is a measure of whether the data are heavy-tailed or light-tailed relative to a normal distribution. That is, data sets with high kurtosis tend to have heavy tails, or outliers.

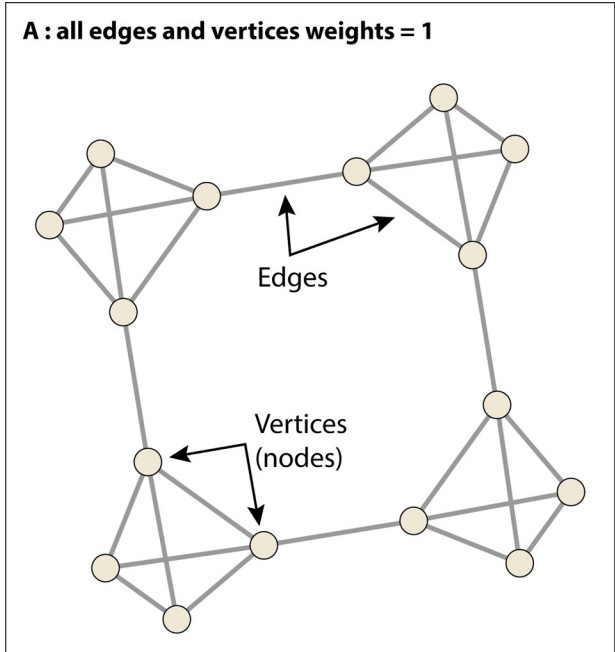
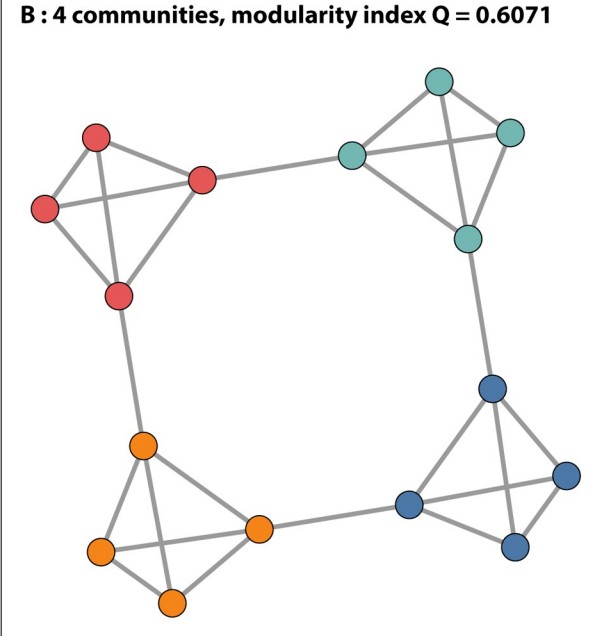
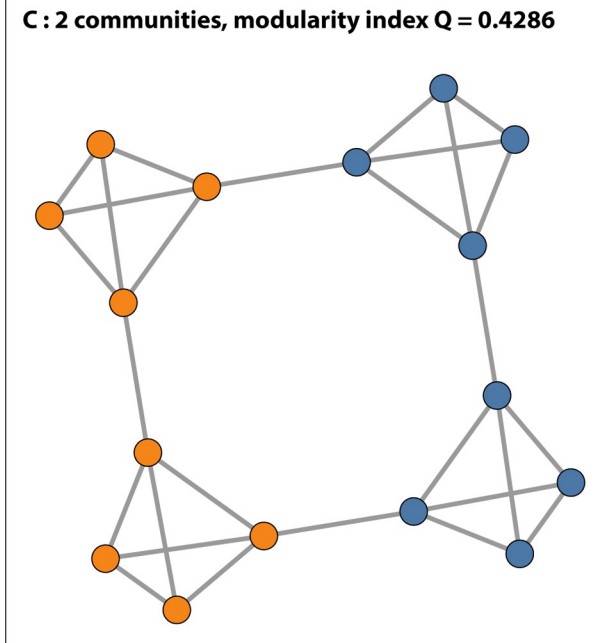
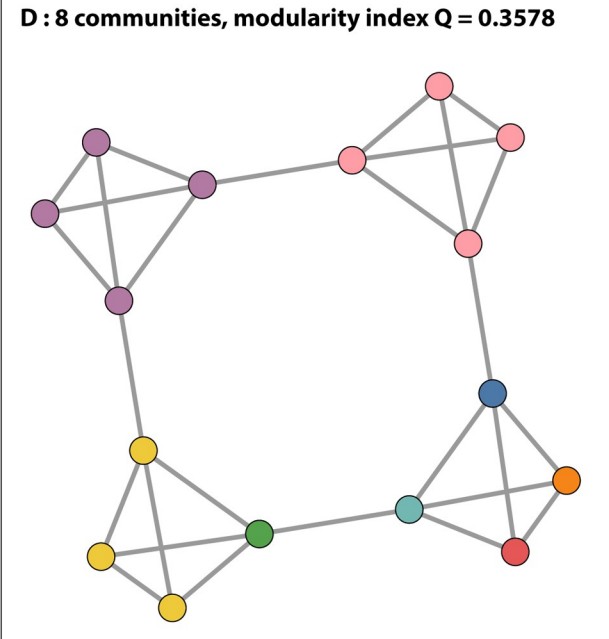

**Fig 1. Illustration of the modularity index.** In this fictitious configuration where 16 vertices form a network (panel A), we try to highlight some subgroups that are naturally communicate with one another. This scenario would naturally lead us to form four communities (panel B). The achieved modularity index is maximum (Q = 0.6071). Conversely, other configurations with 2 or 8 communities (panel C and D) reduce the Q index (0.4286 and 0.3578).

## Comparing 2 clustering methods

Clusters cannot be identified simply by setting a minimum number of patients transferred between dialysis centres per year because the volume of patients handled by the centres varies greatly among regions.

The chosen agglomerative method is the Fast Greedy method and the divisive one is the Edge Betweenness community detection method [9, 10]. The agglomerative Fast Greedy algorithm uses a "gluttonous" optimization. Initially, each health facility is alone. Then it adds to it 1 then 2 etc. other healthcare facilities, thus maximizing the modularity. The algorithm repeats itself until a stopping rule: if the modularity score of iteration n is not increased at iteration n +1, then the algorithm stops; otherwise the clustering continues.

The Edge Betweenness community detection method is divisive in that it starts with the set of healthcare facilities grouped into one unique national cluster, then looks for the edges in the network that are most "between" other facilities, meaning that the edge is, in some sense, responsible for connecting many pairs of other facilities as a bridge does between islands. The algorithm repeat itself until a stopping rule, that is, after the largest increase in modularity estimated by the algorithm.

Python iGraph was used to perform all analysis.

### Validation of the network construction

The validation was based on the opinion of the field actors at the regional level of the REIN registry. The results of the 2 algorithms were illustrated on a map and sent to the 27 regional levels of the REIN registry (coordinating nephrologist and research assistants of the registry) that represented the networks built by our approach and the local perception of network organisation. The actors were asked to choose the most appropriate algorithm. The actors could modify to some degree the grouping of dialysis units.

## Results

At the end of 2018, ESRD care in France took place in 1,166 dialysis units owned by 326 different health care providers (6% public university hospitals, 12% public non-university hospitals, 54% private non-profits, and 27% private for-profit units), and 34 university hospital centres that perform transplantation. During 2016–2018, 32 965 transfers occurred between dialysis units.

With the Edge Betweenness method, the 1,114 French dialysis units in metropolitan France were classified into 156 networks and with the Fast Greedy algorithm, 167 networks (Table 1).

Among the 32 965 transfers, 23 168 (70%) were defined in the same cluster by the Edge Betweenness algorithm and 26 016 (79%) in the same cluster by the Fast Greedy method

**Table 1. Comparison of the Edge Betweenness and Fast Greedy methods for analysing dialysis network activities in France.**

| | Edge Betweenness | Fast Greedy | Fast Greedy correction by field actors |
|---|---|---|---|
| Number of clusters | 156 | 167 | 169 |
| | Transfer within the same network | | |
| Total number of transfers | 23 168 (70.3%) | 26 016 (78.9%) | 25 906 (78.6%) |
| Number of transfers within one network, mean±SD | 170±237 | 158±121 | 156±124 |
| Skeeness | 3.6 | 1.4 | 1.4 |
| Kurtosis | 19.6 | 2.5 | 2.3 |
| Number of transfers within one network, median (IQR) | 102 (31.5–190) | 124 (72–212) | 122 (66–215) |
| | Transfer between different networks | | |
| Total number of transfers | 9 797 (29.7%) | 6 949 (21.1%) | 7 059 (21.4%) |
| Number of transfers between 2 networks, mean±SD | 5.4±16.9 | 3.0±5.5 | 3.0±5.6 |
| Number of transfers between 2 networks, median (IQR) | 1 (1–3) | 1 (1–2) | 1 (1–2) |

IQR, interquartile range

(Table 1). The distribution of the number of transfers within each cluster was narrower with the Fast Greedy than the Edge Betweenness method. The number of transfers between networks was higher with Edge Betweenness algorithm than Fast Greedy method (30% vs 21%), with a higher mean number of transfers between 2 networks.

By comparing the 2 different ways to aggregate the dialysis units, all regions chose the Fast Greedy method. Correction at the regional level resulted in 4 additional networks, deletion of 2 networks by fusion between 2 networks, and 31 units reclassified within 12 networks. Among these 169 networks, the median number of dialysis units in a dialysis network was 5 (interquartile range [IQR] 4–9, range 1–22). The median number of transfers within one cluster was 122 (IQR 66–215) (Table 1).

According to the Fast Greedy method, during the study period, 95% of patients received treatment in only one network. The median number of dialysis units in a dialysis network was 5 (IQR 4–9, range 2–22). Fig 2 shows the Fast Greedy clustering in the Aquitaine region. Nine clusters appear, each consisting of highly connected dialysis centres (i.e., with several patients transferred) (Fig 2 Panel A). At least one dialysis center in each cluster were also connected to a central node: Bordeaux university hospital, the only kidney transplant centre in the region. Fig 2 panel B maps all dialyses centres and clusters in the region. Clusters were geographically homogeneous, with short distances between dialysis centres in the cluster. In densely populated areas, such as Bordeaux, the Fast Greedy approach simplifies the geographical complexity.

As presented in Table 2, networks were variable in their composition. The median number of patients per network in December 2019 was 225 (IQR 131–361). Most networks (137 of 167) included at least one private not-for-profit unit. Only 40 networks had a university unit.

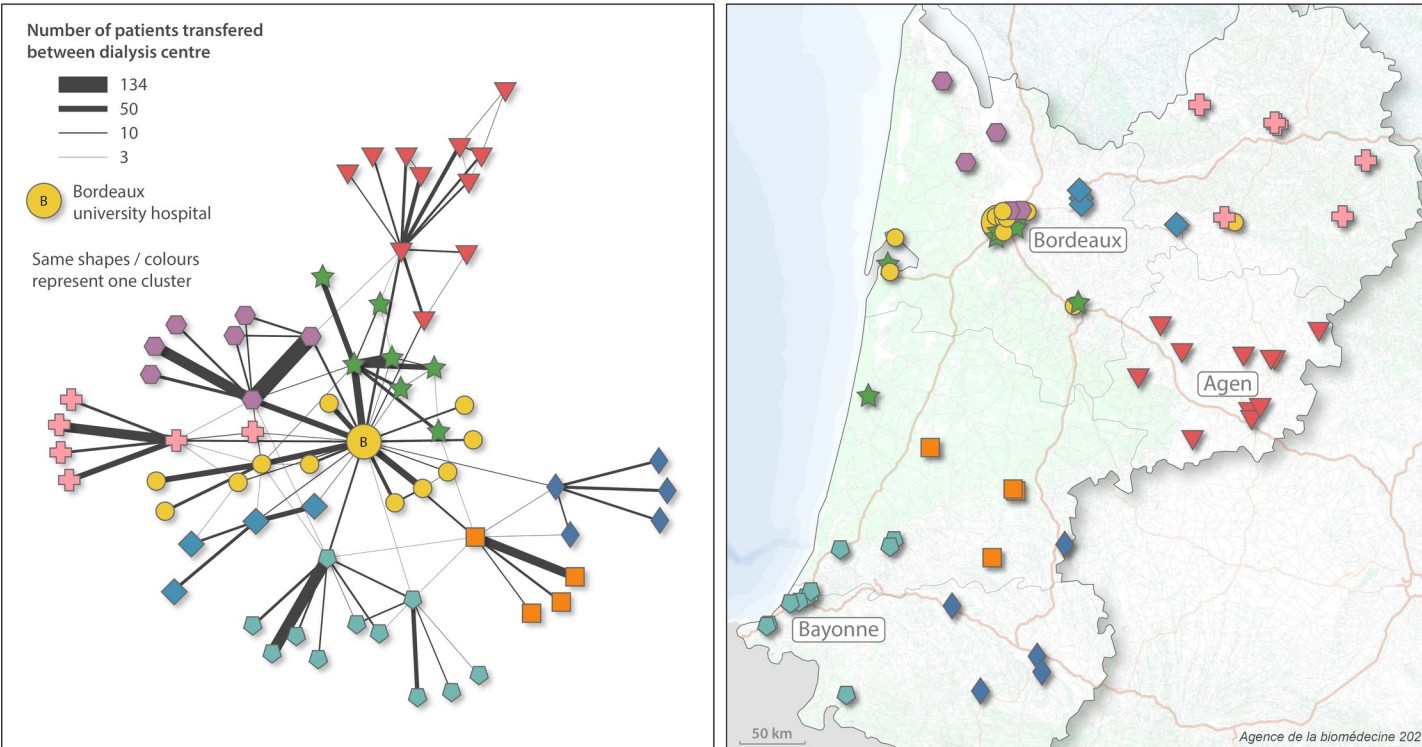

**Fig 2. Results of the Fast Greedy clustering in the Aquitaine region.** The basemap was built from OpenStreetMap open data.: https://www.openstreetmap.org/copyright.

**Table 2. Description of the networks (Fast Greedy method).**

|  | Number of networks concerned | Median (IQR) |
|---|---|---|
| **Number of patients** | 167 | 225 (131–361) |
| **% of patients treated in** |  |  |
| Public units | 92 | 47.5 (29.1–60.9 |
| Private for-profit units | 74 | 71.4 (35.9–100.0) |
| Private not-for-profit units | 137 | 44.0 (31.8–62.3) |
| University units | 40 | 22.5 (16.2–36.3) |
| **% of patients treated by** |  |  |
| In-center HD | 164 | 52.9 (44.3–60.6) |
| Out-center HD | 161 | 25.1 (15.5–33.8) |
| HD self-care unit | 158 | 15.3 (7.4–23.2) |
| Home dialysis (HD or PD) | 142 | 7.4 (3.6–11.0) |

IQR, interquartile range; HD, hemodialysis; PD, peritoneal dialysis

Most of the networks provided the 4 modalities of dialysis treatment, although 25 had no patients who received treatment at home.

## Discussion

There are many methods for detecting communities, and their choice remains complex and, most of the time, empirical [11, 12]. We tested several methods to detect communities and selected the Fast Greedy and Edge Betweenness methods because of their simplicity and calculation speed. According to the opinion of the actors in the field, the Fast Greedy algorithm seemed to be the best method in the context of dialysis activity modelling. The Edge Betweenness classification was not retained because it seemed too sensitive to the volume of links between dialysis units.

### The Fast Greedy method seems well adapted to the dialysis context

According to actors in the field, the Fast Greedy method better reflected the local reality than the Edge Betweenness algorithm. With the former method, the number of classes is defined by the algorithm, but the limitation of this method is that it may group some remote units with larger groups. However, the Fast Greedy method also has several advantages. It does not need a priori knowledge of the number of communities and is one of the fastest algorithms available.

Finding a community structure is a complex task. As part of this study, community numbers and patterns were unknown and of unequal size. The Edge Betweenness algorithm is sensitive to noise with low patient transfer between dialysis centers. Also the number of communities must be chosen manually, for example, by cutting the dendrogram after the largest variance gap.

Our 2 hierarchical classification methods were not constrained by distance. However, this aspect was indirectly included because of the transfer between dialysis units. As well, our hierarchical classification method was not constrained by the volume but was stratified by region to take into account the variability of size, volume and typology of the regions. Therefore, the volume of patients and transfers in each network varied widely. In rural areas, the number of dialysis units and patients are lower than in large urban areas.

We used a functional definition of a network based on the number of transfers between dialysis units. This definition does not correspond to a fixed administrative organization or a

declaration of intent but is based on the observed number of transfers collected in our national registry. In the past, the unit used for quality assessment was usually geographic; for example, the Hospital Service Area is a group of geographic areas served by a single group of health facilities [13]. Our network granularity is smaller than the US ESKD Network Organizations defined geographically by the number and concentration of ESKD beneficiaries in each area. Some US networks represent one state and others multiple states. Our network does not correspond to chain organizations either, which could include various providers [14].

## Theoretical validation

The Fast Greedy and Edge Betweenness algorithms are easily understandable and are recommended as a first approach in detecting communities. Both have different purposes. The Edge Betweenness is a top-down algorithm that focuses more on differences between clusters (interclass variance) and is more sensitive to large volume exchanges, whereas the Fast Greedy is a bottom-up algorithm more discriminating on intraclass variance and based on proportions rather than frequencies.

The 2 approaches give similar results overall but showed significant differences as soon as the volumes treated increase; particularly in the most populated regions of France, such as the Paris region (18% of the French population). The Edge Betweenness method is less discriminating in this context because of its sensitivity to the frequency of the flows, unlike the Fast Greedy method, which takes into account the heterogeneity of dialysis units and region size.

Our interpretation is that the Fast Greedy method, by clumping the entities from close to close, would be a proxy describing which basic components of our network, the dialysis centres, work frequently together. In other words, the aggregation of related fundamental units forms clusters. This is probably the best way to describe the dialysis healthcare offer in France: planning is at local and regional levels according to the population needs in each territory. French dialysis centres each have their own specificities (autonomous patients or not, breaking up enclosed territory or reference centres for patients with more comorbidities, hospital-based or self-care units etc.). Because of the complementarity of units, patients are transferred between the different dialysis units according to their needs and choice, although the choice is mainly according to proximity to home. The Edge Betweenness algorithm, which starts from the most important connections to separate them, seems more adapted to identify reference centres. Its utility to describe a national healthcare organisation such as organ transplantation could be discussed.

## A multiscalar approach to assess quality of care

Quality of care can be analysed at the level of a care unit, for example, if the quality criterion concerns the causal pathway of some complications. However, if the quality criterion is more global, such as for survival time, the whole chain of care for the patient should be considered. The geographical unit seems appropriate when the possible causes of a dysfunction are attached to a territory. For example, this is the case when studying the possible impact of social deprivation on patient care [15] and also when the quality factors studied concern the territorial organization of care. However, access to transplantation, for example, involves all the professionals at different stages. The network approach has been used to explore variability in this practice [16]. As well, because home dialysis and self-care units are mainly provided by not-for-profit units, these patients are transferred to these units for chronic care and are transferred back to public or private for-profit hospital-based dialysis units if necessary (i.e., with worsening of the clinical condition) [17].

Because the diversity of the healthcare offer in a territory, variability of dialysis activities within a network is expected, each unit fulfilling its role in the chain of care. However, this diversity aims to meet patients' needs in a complementary manner and does not represent variability in a treatment policy. Indeed, these units share common procedures and organize multidisciplinary meetings; they even exchange staff. In contrast, between-network variability may reflect variability in a treatment policy and procedures. Therefore, we must analyse whether this variability between networks represents a loss of chance for the patients.

As part of an implementation of a quality care action, our approach can be used to identify networks that statistically differ from the other in a given outcome [16]. It can also be used to detect healthcare facilities with a high centrality score as a "bridgehead" for action. Finally, by identifying "isolated" healthcare facilities, it may help in finding innovative ways of collaboration for the benefit of the patients.

## Strengths and limitations

The strength of this approach is that it is based on observed transfer activities and not administrative representation. It is a fast method, with results that can be updated each year with observed transfers from the previous year. Given the small changes asked by the region, for epidemiological purposes, the automatic classification may be valuable as a first approximation without the need for manual correction. This conclusion has been validated by field actors and the scientific committee of the REIN registry. It is a compromise between a useful level for action and homogeneity. This classification is now used in various French studies exploring variability in care practice [16].

However, this study has some limitations. Our method of network classification was constrained by the fact that one dialysis unit could be only classified in one network and only in a network of its own region. This was decided because of the regional organization of healthcare and the association between local practice and regional level of academic training. However, the final validation by regions resulted in only minor remarks on this automatic classification. Second, the Fast Greedy and Edge Betweenness algorithms aim to optimize the modularity criteria with different algorithms. However, an optimal community structure algorithm with the aim of finding the maximum modularity score could be used. This algorithm would test all possible partitions but has the main disadvantage of being a NP-complex problem, with a time exponential solving complexity. Third, the criterion to define a collaboration were based on a patient's transfer between 2 dialysis units and not a global strategy throughout the trajectory of care. This situation does not directly imply the nature of the relationship in terms of practices. A distinction could be made between communities with shared practices and those that provide follow-up for the same patients. To answer this question, more in-depth information concerning protocols and quality assurance programmes in each unit needs to be collected. These communities with shared practice may collaborate throughout the country because they provide the same modality of treatment or have a similar case mix of patients, (e.g., home dialysis units or university hospital-based hemodialysis units). According to the criteria of evaluation, one or more of these clusters may be relevant. To evaluate the strategy of care in the trajectory of the patients, communities that provide follow-up for the same cohort should be preferred.

## Conclusions

The Fast Greedy method is a relative simple method to aggregate dialysis units when a "collaborative work" indicator is available. We chose transfers between units as an indicator because it represents a real life activity and the data were available in our registry. This forward agglomerative method seems more appropriate than a divisive method to model dialysis healthcare

owing to the complementarity between dialysis units and organization at a local level to ensure proximity treatment.

## Acknowledgments

The authors thank all the REIN registry participants, especially nephrologists and professionals in charge of data collection and quality control. Dialysis and transplant centres participating in the registry are listed in the REIN annual report (https://www.agence-biomedecine.fr/Les-chiffres-du-R-E-I-N).

## Author Contributions

**Conceptualization:** Cécile Couchoud, René Ecochard, Mathilde Prezelin-Reydit, Thierry Lobbedez, Florian Bayer.

**Formal analysis:** Cécile Couchoud, Florian Bayer.

**Methodology:** Cécile Couchoud, René Ecochard, Florian Bayer.

**Software:** Florian Bayer.

**Supervision:** Cécile Couchoud.

**Validation:** Cécile Couchoud, René Ecochard, Mathilde Prezelin-Reydit, Thierry Lobbedez, Florian Bayer.

**Writing – original draft:** Cécile Couchoud, Florian Bayer.

**Writing – review & editing:** Cécile Couchoud, René Ecochard, Mathilde Prezelin-Reydit, Thierry Lobbedez, Florian Bayer.

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
