## [Decision Letter · Decision Letter 0]

7 Sep 2022

PONE-D-22-06137Functional representation of the network organisation of dialysis activities in France: a novel level for assessing quality of carePLOS ONE

Dear Dr. Couchoud,

Thank you for submitting your manuscript to PLOS ONE. After careful consideration, we feel that it has merit but does not fully meet PLOS ONE’s publication criteria as it currently stands. Therefore, we invite you to submit a revised version of the manuscript that addresses the points raised during the review process.

We look forward to receiving your revised manuscript.

Kind regards,

Pierre Delanaye

Academic Editor

PLOS ONE

Journal Requirements:

3. We note that Figure 1 in your submission contain map/satellite image which may be copyrighted. All PLOS content is published under the Creative Commons Attribution License (CC BY 4.0), which means that the manuscript, images, and Supporting Information files will be freely available online, and any third party is permitted to access, download, copy, distribute, and use these materials in any way, even commercially, with proper attribution. For these reasons, we cannot publish previously copyrighted maps or satellite images created using proprietary data, such as Google software (Google Maps, Street View, and Earth). For more information, see our copyright guidelines: http://journals.plos.org/plosone/s/licenses-and-copyright.

Additional Editor Comments:

Please answer tot the comments of the reviewers.

Reviewers' comments:

Reviewer's Responses to Questions

**Comments to the Author**

1. Is the manuscript technically sound, and do the data support the conclusions?

Reviewer #1: Yes

Reviewer #2: Yes

2. Has the statistical analysis been performed appropriately and rigorously? 

Reviewer #1: Yes

Reviewer #2: Yes

3. Have the authors made all data underlying the findings in their manuscript fully available?

Reviewer #1: Yes

Reviewer #2: No

4. Is the manuscript presented in an intelligible fashion and written in standard English?

Reviewer #1: Yes

Reviewer #2: Yes

5. Review Comments to the Author

Reviewer #1: This article by Couchoud et al tries to identify networks of dialysis units in France by 2 different methods to describe patient pathway according to their health status and home localization, reflecting real life.

Article is well written and easily understandable. It will be cited in all future article using this methodology.

This article presents an original approach of dialysis offer and patient transfers. It also show the diversity of the networks according probably to the offer, the geography and the medical uses. French organization of health is best explain by this clustering and might be different from others countries where they might be limitation of transfer because of health system.

Validation of the best method by the actor of the field is an pragmatic way to control the work.

This clustering will allow to best compare patient pathways between two regions/networks than comparation between centers since demographical characteristics might be very different between centers not allowing direct comparison while networks including all dialysis modalities should be more comparable.

Minor correction:

Page 8, ligne 11: “whe distribution” instead of “The”

Reviewer #2: Thank you for giving me the opportunity to review the manuscript by Couchoud et al about the representation of the network organization of dialysis activities in France.

In this manuscript, the authors aim to go deeper in the understanding of center effect, accounting for the specific organization of dialysis activities in France. They focus on the specific organization of the French dialysis network, but the spirit of this work could/should inspire other teams to conduct similar evaluations in their own country based on their specific organization.

Of course, French readers will have a specific interest in reading this manuscript, but readers from others country will certainly be quite interested in this text.

The article is clear although it deals with theoretical, sometimes complex concepts.

I have very few comments, the purpose of which is mainly to suggest points that will help improve an already very good manuscript

1- In the Method part, the authors state that there is more than 1150 dialysis units in France, with a maximum number of patients of 320, could they also state the minimum number of patients?

2- Is there some published source about the time to a dialysis unit for patients (17 minutes)?

3- I applaud the effort of the authors to explain clearly the terms used: degree, index, closeness centrality, betweenness. This would even be easier to get for the reader with a dedicated figure (in supplemental material if necessary) with examples on drawn networks.

Minor points:

Typo in page 8: “Whe distribution of the number of transfers within each cluster was narrower ... » : “whe” should be replaced by “the”.

The format of the references is not uniform and should be therefore modified.

“Skeeness” and “kurtosis” used in table 1 should be defined in the “vocabulary” paragraph or in the legend of table 1.

6. PLOS authors have the option to publish the peer review history of their article (what does this mean?). If published, this will include your full peer review and any attached files.

Reviewer #1: No

Reviewer #2: No

---

## [Author Response · Author response to Decision Letter 0]

16 Sep 2022

Point-by-point answer to reviewers.

Reviewer 1

“This article by Couchoud et al tries to identify networks of dialysis units in France by 2 different methods to describe patient pathway according to their health status and home localization, reflecting real life.

Article is well written and easily understandable. It will be cited in all future article using this methodology.

This article presents an original approach of dialysis offer and patient transfers. It also show the diversity of the networks according probably to the offer, the geography and the medical uses. French organization of health is best explain by this clustering and might be different from others countries where they might be limitation of transfer because of health system.

Validation of the best method by the actor of the field is an pragmatic way to control the work. This clustering will allow to best compare patient pathways between two regions/networks than comparation between centers since demographical characteristics might be very different between centers not allowing direct comparison while networks including all dialysis modalities should be more comparable.”

Thank you for your positive comments.

We have corrected the typo: Page 8, ligne 11: “whe distribution” instead of “The”.

Reviewer 2

Thank you for giving me the opportunity to review the manuscript by Couchoud et al about the representation of the network organization of dialysis activities in France.

In this manuscript, the authors aim to go deeper in the understanding of center effect, accounting for the specific organization of dialysis activities in France. They focus on the specific organization of the French dialysis network, but the spirit of this work could/should inspire other teams to conduct similar evaluations in their own country based on their specific organization.

Of course, French readers will have a specific interest in reading this manuscript, but readers from others country will certainly be quite interested in this text.

The article is clear although it deals with theoretical, sometimes complex concepts.

I have very few comments, the purpose of which is mainly to suggest points that will help improve an already very good manuscript.

Thank you for your positive comments.

1- In the Method part, the authors state that there is more than 1150 dialysis units in France, with a maximum number of patients of 320, could they also state the minimum number of patients?

Very small self-care units have only 2 patients. This information was added.

2- Is there some published source about the time to a dialysis unit for patients (17 minutes)?

We didn’t published a specific study on this matter. This information is calculated each year and used in various studies.

2- I applaud the effort of the authors to explain clearly the terms used: degree, index, closeness centrality, betweenness. This would even be easier to get for the reader with a dedicated figure (in supplemental material if necessary) with examples on drawn networks.

We added an additional figure as suggested.

Minor points:

Typo in page 8: “Whe distribution of the number of transfers within each cluster was narrower ... » : “whe” should be replaced by “the”.

Done. Thank you.

The format of the references is not uniform and should be therefore modified.

The difficulty is that most of the social science publications are published on website. We have reformatted some references.

 “Skeeness” and “kurtosis” used in table 1 should be defined in the “vocabulary” paragraph or in the legend of table 1.

Done. Thank you for the suggestion.

---

## [Editor Report · Decision Letter 1]

28 Sep 2022

Functional representation of the network organisation of dialysis activities in France: a novel level for assessing quality of care

PONE-D-22-06137R1

Dear Dr. Couchoud,

We’re pleased to inform you that your manuscript has been judged scientifically suitable for publication and will be formally accepted for publication once it meets all outstanding technical requirements.

Kind regards,

Pierre Delanaye

Academic Editor

PLOS ONE

Additional Editor Comments (optional):

No further comments
---

## [Editor Report · Acceptance letter]

11 Oct 2022

PONE-D-22-06137R1 

Functional representation of the network organisation of dialysis activities in France: a novel level for assessing quality of care 

Dear Dr. Couchoud:

I'm pleased to inform you that your manuscript has been deemed suitable for publication in PLOS ONE. Congratulations! Your manuscript is now with our production department. 

Kind regards, 

on behalf of

Professor Pierre Delanaye 

Academic Editor

PLOS ONE